# Changes in enzyme activity and microbial community of rhizosphere soil under continuously monocultured *Passiflora edulis* treatment

Weiwei Lin[1☯‡], Zhihan Chen[2☯‡], Zhaowei Li[3*], Wenxiong Lin[2*]

**1** College of Food Engineering, Zhangzhou Institute of Technology, Zhangzhou, Fujian, China, **2** College of Agriculture, Fujian Agriculture and Forestry University, Fuzhou, Fujian, China, **3** College of Juncao Science and Ecoogy, Fujian Agriculture and Forestry University, Fuzhou, Fujian, China

☯ These authors contributed equally to this work.
‡ These authors shares co-first authors on this work.
* lwx@fafu.edu.cn (WL); zhaoweili@fafu.edu.cn (ZL)

## Abstract

In response to the severe continuous cropping obstacles encountered during the cultivation of passion fruit, which leads to significant declines in yield and quality, as well as rampant pests and diseases, it is particularly important to explore strategies for mitigating these obstacles. The present study used the rhizosphere soil samples from one-year-old (FY, first-year cropping) and two-year-old (SY, second-year cropping) "Golden Passion Fruit" plants, along with soil from uncropped land (CK) as a control and the techniques such as high-throughput sequencing, qRT-PCR, and HPLC-MS to analyze the main physicochemical properties, phenolic acid content, and microbial community changes in the rhizosphere soil of passion fruit under different continuous cropping durations, the results indicated that the contents of total nitrogen, total phosphorus, available nitrogen, available phosphorus, available potassium, and organic matter in FY soil were significantly higher than those in SY soil, and the pH value of the FY soil was also significantly higher than that of SY soil. Additionally, compared with FY soil, the activities of polyphenol oxidase, peroxidase, urease, and invertase in SY soil were significantly reduced by 18.0%, 43.6%, 19.8%, and 45.5%, respectively. HPLC analysis revealed that the concentrations of syringic acid, vanillin, benzoic acid, and ferulic acid in the SY soil were significantly increased by 18.0%, 21.9%, 24.4%, and 21.1%, respectively, compared to those in the FY soil. qRT-PCR analysis showed that as the duration of continuous cropping increased, the total number of bacteria in the rhizosphere soil of passion fruit decreased by 9.37%, while the total number of fungi increased by 57.8%. High-throughput sequencing results demonstrated that at the genus level, the relative abundances of Acidothermus, Acidibacter, Bacillus, and Acidobacterium were significantly increased by38.0%, 56.3%, 34.3%, 77.3%. whereas the relative abundances of Rhizomicrobium, Nitrospira,

**Data availability statement:** All relevant data are within the manuscript and its Supporting information files.

**Funding:** This study supported by the Doctoral Research Start-up Fund Project of Zhangzhou Institute of Technology in 2024 (Project No.: ZZYB2403), the School-level Project of Zhangzhou Institute of Technology in 2024 (zzykyk24004), and the Natural Science Foundation of Fujian Province (2022J01142). The funders had no role in study design, data collection and analysis, decision to publish, or preparation of the manuscript.

**Competing interests:** The authors have declared that no competing interests exist.

Burkholderia, Sphingomonas, Gemmatimonas, Streptomyces, and Nocardioides were significantly lower 45.2%,59.3%,50.6%,89.1%,74.5%,82.7% in the SY soil, relative to those in the FY soil. In summary, as the duration of continuous cropping increases, the soil fertility, enzyme activity, pH value, and beneficial microbial content in the rhizosphere of passion fruit decrease significantly, while the contents of phenolic acids and pathogenic microorganisms increase significantly. The findings of this study provide a theoretical basis for further elucidating the formation mechanism and mitigation strategies of continuous cropping obstacles in passion fruit.

## 1. Introduction

Passion fruit, native to North America, is a plant belonging to the *Passifloraceae* family and the *Passiflora* genus. It holds significant socio-economic value in both food and medicinal applications and is widely used in fresh fruit, beverages, and spices in countries like Brazil, Colombia, Vietnam, and regions such as Taiwan, China [1,2]. Passion fruit was introduced to mainland China in the 1980s. Currently, its cultivation area in southern provinces of China such as Fujian, Guangxi, Hainan, Guangdong, and Yunnan has exceeded 40,000 hectares [3]. Due to its characteristic of yielding results in the same year of planting, passion fruit has played a crucial role in poverty alleviation in recent years. As a perennial vine plant, passion fruit has a high yield in the first year of planting. However, from the second year onwards, the yield gradually decreases, and the frequency of pest and disease outbreaks increases, leading to a decline in both yield and quality. How to mitigate or eliminate the continuous cropping obstacles in passion fruit has become a cutting-edge research topic among domestic and international peers [4].

Previous studies have shown that the causes of continuous cropping obstacles in plants include the degradation of soil physicochemical properties, depletion of nutrient status [5,6], allelopathic autotoxicity of root exudates [7,8], and the deterioration of the soil microenvironment, such as a decrease in microbial diversity, an increase in pathogenic microorganisms, and exacerbation of pests and diseases [9]. To address declining soil fertility, people commonly adopt measures such as increasing fertilizer application. However, the effect on alleviating continuous cropping obstacles is not significant, and it even exacerbates environmental pollution [10,11]. For instance, if the main grain crop in the North, wheat, is continuously cropped, its yield will drop by more than half, even with sufficient chemical fertilizers. Similarly, Liang et al. [12] also conducted research on preventing cucumber continuous cropping obstacles by increasing fertilization, finding that the decline in soil nutrients is not the main cause of continuous cropping obstacles, indicating differing views on whether nutrient decline in soil is the reason for continuous cropping obstacles. In continuous cropping scenarios, the nutrient content in the soil does not decrease with the increase of continuous cropping years. This may be due to the phenolic acids exuded by continuous cropping plants into the rhizosphere, which are allelopathic substances in the soil environment, significantly inhibiting continuously monocultured crop growth

and development, causing soil metabolic imbalance, and soil nutrient sequestration, leading to poor plant growth and development [13]. Therefore, the causes of plant continuous cropping obstacles formation and exacerbation are complex and diverse, with various factors interconnecting and interacting, resulting from the comprehensive interaction of multiple factors within the plant and soil systems.

Brazil and south China are the world's largest producer of yellow passion fruit,YPF, with an annual production of more than 714000 ton. Other species, including purple passion fruit, account for less than 5% of the area of passion fruit cultivation, indicating their limited commercialization in Brazil and China. However, the area expansion of passion flower monoculture was followed by the emergence and worsening of a large number of diseases, which was caused by continuously monocropping practice [14,15]. Currently, research on continuous cropping of passion fruit is limited. Only Wang et al. [16] have studied the autotoxicity of continuous cropping soil of passion fruit, Chen et al. [17] and Zou et al. [18] have quantitatively studied the microbial flora and biomass in the continuous cropping soil of passion fruit. The results showed that with the increase in continuous cropping years of passion fruit, the number of soil bacteria and actinomycetes decreases significantly, while the number of fungi increases significantly. However, the interaction mechanism among the three factors of "plant-soil-microorganisms" in the rhizosphere process has not been deeply studied [19]. Accordingly, this study focused on the rhizosphere soil of 'Yellow Passion Fruit' that had undergone continuous cultivation for two years as the research subject. The rhizosphere soil from plants cultivated for one year and blank soil were used as controls. Utilizing conventional soil chemical analysis methods and high-throughput sequencing technology, this research thoroughly examined the characteristics of rhizosphere chemical and microbial ecological changes induced by continuous cropping, which contribute to the occurrence of continuous cropping obstacles in Yellow Passion Fruit plants. The objective is to provide a foundation for further investigations into mitigating continuous cropping obstacles and promoting sustainable development within the passion fruit industry.

## 2.  Materials and methods

### 2.1.  Experiment location and test materials

The trial site was located in Jiyang Town, Jian'ou City, Fujian Province, southeast China, serving as an off-campus experimental base for the Institute of Agricultural Ecology at Fujian Agriculture and Forestry University (27°11′N-118°12′E). The average altitude ranged from 130 to 1383 meters, with an annual average temperature of 18.1°C and an annual average precipitation of 1557–1743 mm. The soil type was sandy loam, exhibiting acidic pH, and moderate organic matter content. The passion fruit variety cultivated in the trial area was "Yellow Passion Fruit." The experiment was conducted from 2022 to 2024 for positional observation, setting up plots with passion fruit planted for 1 year as FY (first-year crop), 2 years as SY (second-year crop), and adjacent soil without passion fruit as CK (control). The orchard featured similar environmental characteristics in terms of height, slope position, and slope gradient. The planting time was around March each year, with a row spacing of 2m × 3m. There were about 16800 plants are planted per hectare. Dig planting holes with a length, width, and depth of 60 cm each during planting, and apply 3–5 kg of organic fertilizer, 0.5 kg of calcium magnesium phosphate fertilizer, and 0.5 kg of biological bacterial fertilizer to each hole. After harvesting passion fruit, it is necessary to promptly remove old and overly dense leaves, prune excess non fruiting branches, diseased, old and weak branches, etc., to avoid excessive absorption of nutrients. The field management measures for experimental plots with different continuous cropping years are consistent. Each treatment in the experimental orchard consisted of one plot with three replicates, each plot covering 300 m². The annual harvest time for passion fruit was from September to December. On July 10, 2024, during the fruit enlargement period of passion fruit, root zone soil samples from FY, SY, and CK were collected using the five-point method, with three replicates per treatment. The collected soil samples were sieved through a 2 mm mesh to remove impurities, immediately stored in an ultra-low temperature freezer at −80°C for subsequent sequencing; another portion was used for soil physicochemical properties.

## 2.2. Soil chemical property determination

The pH value was determined by employing the composite electrode method, with a soil-to-water ratio of 1:2.5. Potassium content was measured using a flame photometer. The measurement and preparation of soil total nitrogen, available nitrogen, total phosphorus, available phosphorus, total potassium, and available potassium followed the protocols described by Lu Rukun [20].

## 2.3. Determination of soil enzyme activity

Soil enzyme activities were measured according to the methods described by Zhou Likai and colleagues [21]. The urease activity was determined using the phenol sodium colorimetric method. Sucrase activity was assessed using the 3,5-dinitrosalicylic acid colorimetric method. Catalase activity was measured using the potassium permanganate titration method. Polyphenoloxidase activity was determined using the pyrogallol colorimetric method.

## 2.4. Quantitative real-time polymerase chain reaction (qRT-PCR) analysis of key microorganisms in the rhizosphere soil

The total soil DNA was extracted using the BioFast Soil Genomic DNA Extraction Kit provided by Hangzhou Borui. Subsequently, the concentration of the extracted total soil DNA was measured using the Nanodrop 2000C instrument. Based on the obtained results, the total populations of bacteria and fungi in the rhizosphere soil were analyzed. Bacterial quantification was performed using the Eub338/Eub518 primers, while fungal quantification utilized the ITS1F/ITS4 primers. Detailed methods can be found in references [22,23].

## 2.5. Extraction of phenolic acids from rhizosphere soil

The ethyl acetate extraction method [19,23], was utilized for phenolic acid extraction. In detail, each sample was treated with 40 mL of 1 M sodium hydroxide followed by 90 minutes of sonication. Subsequently, the suspension was centrifuged at 10,000 rpm for 10 minutes, and the supernatant was collected. The filtrate was then adjusted to a pH of 2.5 using 9 M hydrochloric acid, and another centrifugation step at 10,000 rpm for 10 minutes was performed to collect the supernatant. Ethyl acetate extraction was repeated five times, and the combined extract was freeze-dried at low temperature. The resulting extract was resuspended in 5 mL of methanol, yielding the phenolic acid extraction solution, which was stored at 4°C in the dark. The collected phenolic acid extraction solution was filtered through a 0.22μm membrane filter and subjected to HPLC-MS analysis. The chromatographic conditions included a mobile phase A of 2% acetic acid and a mobile phase B of acetonitrile, with a flow rate of 0.7 mL/min. The injection volume was 10 μL, and a gradient elution program was employed: A: 95% (0 min) → 90% (10 min) → 88% (30 min) → 72% (34 min) → 72% (40 min) → 95% (41 min) → 95% (43 min). Mass spectrometry parameters included an IonSource of ESI, Scan Mode set to Negative Ion Mode, Vaporizer Temperature at 350°C, Capillary Temperature at 350°C, and Capillary Voltage of −35 V. Qualitative and quantitative analysis was performed by comparing the retention time with that of the standard substance. Furthermore, the collected phenolic acid extraction solution was also filtered through a 0.22 μm membrane filter for high-performance liquid chromatography (HPLC) analysis using a C18 column (Inertsil ODS-SP, 4.6 × 250 mm, 5 μm). The chromatographic conditions were as follows: mobile phase A was methanol, mobile phase B was 2% acetic acid, with a flow rate of 0.7 mL/min. Detection was conducted at a wavelength of 280 nm, and the injection volume was set to 20 μL. The column temperature was maintained at 30°C. Qualitative analysis was performed by comparing the retention time with that of the standard substance, while quantitative analysis was based on peak area measurements.

## 2.6. Extraction of rhizosphere soil DNA and PCR amplification

The DNA extraction method for rhizosphere soil referred to the instructions provided by the DNA Kit (Omega Bio-tek, Norcross, GA, U.S.). Three samples of rhizosphere soil DNA were extracted, and their quality was assessed through 1%

agarose gel electrophoresis. The genomic DNA was then diluted with sterile water to a concentration of 1 ng/μL. Subsequently, using the diluted DNA as a template, amplification of the bacterial 16S rDNA V3-V4 region was performed using PerfectStart Green qPCR SuperMix and a high-fidelity enzyme (New England Biolabs, USA). The primers used were 338F (5'-ACTCCTACGGGAGGCAGCA-3') and 806R (5-GGACTACHVGGGTWT-3') [19].

For fungal analysis, the fungal ITS1 region was amplifiedusingtheprimers ITS1(5'-CTTGGTCATTTAGAGGAAGTAA-3') and ITS2(5'-TGCGTTCTTCATCGATGC-3') [24]. The PCR reaction conditions were as follows: initial denaturation at 95°C for 5 minutes, denaturation at 95°C for 45 seconds, annealing at 55°C for 50 seconds, extension at 72°C for 45 seconds, for a total of 32 cycles, followed by a final extension at 72°C for 10 minutes. The PCR products were sent to Beijing Orison Genetics Technology Co., Ltd. for further analysis and determination.

## 2.7. Data analysis

After undergoing quality control with the FastQC software, the raw sequencing data was subjected to further processing using the Cutadapt software (V1.9.1) to eliminate short sequences (<200 bp) and low-quality sequences (q < 25) [25]. Operational Taxonomic Units (OTUs) were assigned to the remaining sequences, which exhibited a 97% similarity, employing the Mothur method in combination with the SILVA database. Based on the taxonomic information, bar charts illustrating species classification and heatmaps showcasing species abundance were generated. The coverage and diversity indices of the samples were calculated using the QIIME software. To gain insights into the functional and metabolic pathways of the microbial community, PICRUSt software was employed for predictive analysis [26]. Graphical representations were created utilizing GraphPad Prism 9 and Origin 2019 software. For data organization, Microsoft Excel 2013 was employed, while statistical analysis was performed using DPS 7.05 and SSPS 19.0 software. The intergroup differences were tested using the LSD method.

## 3. Results

### 3.1. Effects of continuous monoculture on the physiochemical properties of rhizosphere soil in passion fruit

As indicated in Table 1, Compared with FY treatment, SY treatment significantly reduced the total nitrogen, total phosphorus, available nitrogen, available phosphorus, available potassium, and organic matter content in the rhizosphere soil of passion fruit, with no significant difference in total potassium content. However, the CK treatment significantly reduced the content of total nitrogen, total phosphorus, total potassium, available nitrogen, available phosphorus, available potassium, and organic matter. Compared with the CK treatment, the SY treatment significantly increased the content of total nitrogen, total potassium, available nitrogen, available phosphorus, available potassium, and organic matter in the rhizosphere

**Table 1. Basic physicochemical properties of rhizosphere soil of continuous cropping passion fruit.**

| Soil physicochemical properties | CK | FY | SY |
|---|---|---|---|
| Total nitrogen(TN)(g/kg) | 0.73c | 2.40a | 0.86b |
| Total phosphorus(TP)(g/kg) | 0.65b | 0.96a | 0.67b |
| Total potassium(TK)(g/kg) | 0.72b | 0.88a | 0.89a |
| Available nitrogen(AN)(g/kg) | 0.50c | 1.12a | 0.67b |
| Available phosphorus(AP)(mg/kg) | 0.67b | 2.16a | 0.68b |
| Available potassium(AK)(mg/kg) | 7.66c | 12.30a | 9.27b |
| Soil organic matter(SOM)(g/kg) | 160.14c | 199.21a | 169.15b |
| pH | 5.15a | 4.09b | 3.85c |

Note: FY (first consecutive monoculture years); SY (second consecutive monoculture years). CK (control soil). Data with different lowercase letters on same column indicate significant different at P < 0.05.

soil of passion fruit, except for no significant difference in total phosphorus and available phosphorus. As the years of continuous cropping of passion fruit increased, the pH value also decreased; the CK treatment had the highest pH value, and the SY treatment had the lowest pH value. It is evident that continuous monoculturre of passion fruit led to a significant decline in soil fertility and pH value in the rhizosphere.

### 3.2. Changes in phenolic acid content in rhizosphere soil of passion fruit under continuous monoculture regime

As shown in Table 2, compared with the FY treatment, the contents of vanillin, benzoic acid, syringic acid, and ferulic acid in the rhizosphere soil of passion fruit under the SY treatment increased significantly. In contrast, the contents of these phenolic acids in the rhizosphere soil of passion fruit under the CK treatment decreased significantly. It is evident that the overall trend in phenolic acid content in the rhizosphere soil under continuous monocropping follows CK < FY < SY. This finding suggests a complex interplay between continuous cropping and changes in phenolic acid profiles, which may influence soil health and plant growth.

### 3.3. Effect of continuous cropping on enzyme activities in rhizosphere soil of passion fruit

Soil enzymes play crucial roles in various biochemical processes within the soil, and their activities directly reflect the strength of the soil's biochemical properties. Significant differences in enzyme activities were observed in the rhizosphere soil of passion fruit under different cropping years, as depicted in (Fig 1, S1 Table). Compared to the FY treatment, the SY and CK treatments showed significantly decreased activities of peroxidase, polyphenoloxidase, urease, and sucrase in the rhizosphere soil, and there was no significant difference between the SY and CK treatments. This indicates that as the years of continuous monocropping of passion fruit increase, the activities of soil enzymes in the rhizosphere soil significantly decline. This reduction can have profound implications for soil health and the overall productivity of the passion fruit crop.

### 3.4. Effect of continuous monocropping on bacterial community diversity and structure in rhizosphere soil of passion fruit crop

The species richness and evenness of bacterial communities in the rhizosphere soil of continuously – monocropped passion fruit were evaluated. As shown in Table 3, compared with the FY treatment, the SY treatment had significantly lower values in the Chao1 index, Observed species index and Shannon index for bacteria. The CK treatment had significantly higher values in the Observed species index and Shannon index, while there was no significant difference in the Simpson index. It is evident that as the years of continuous monocropping of passion fruit increase, the richness and evenness of bacterial communities in the rhizosphere soil significantly decline. These changes can impact soil microbial ecology and may affect soil health and plant growth dynamics.

Table 2. Variation in the content of phenolic acids in rhizosphere soil of continuously monocultured passion fruit plants.

| Treatment | Syringic acid content (ug/g) | Vanillin content (ug/g) | Benzoic acid content (ug/g) | Ferulic acid content (ug/g) |
|---|---|---|---|---|
| CK | 59.17c | 41.17c | 28.63c | 50.65c |
| FY | 91.89b | 52.49b | 38.04b | 85.65b |
| SY | 112.12a | 67.18a | 50.33a | 108.56a |

Note: FY (first consecutive monoculture years); SY (second consecutive monoculture years). CK (control soil).Data with different lowercase letters on same column indicate significant different at P < 0.05.

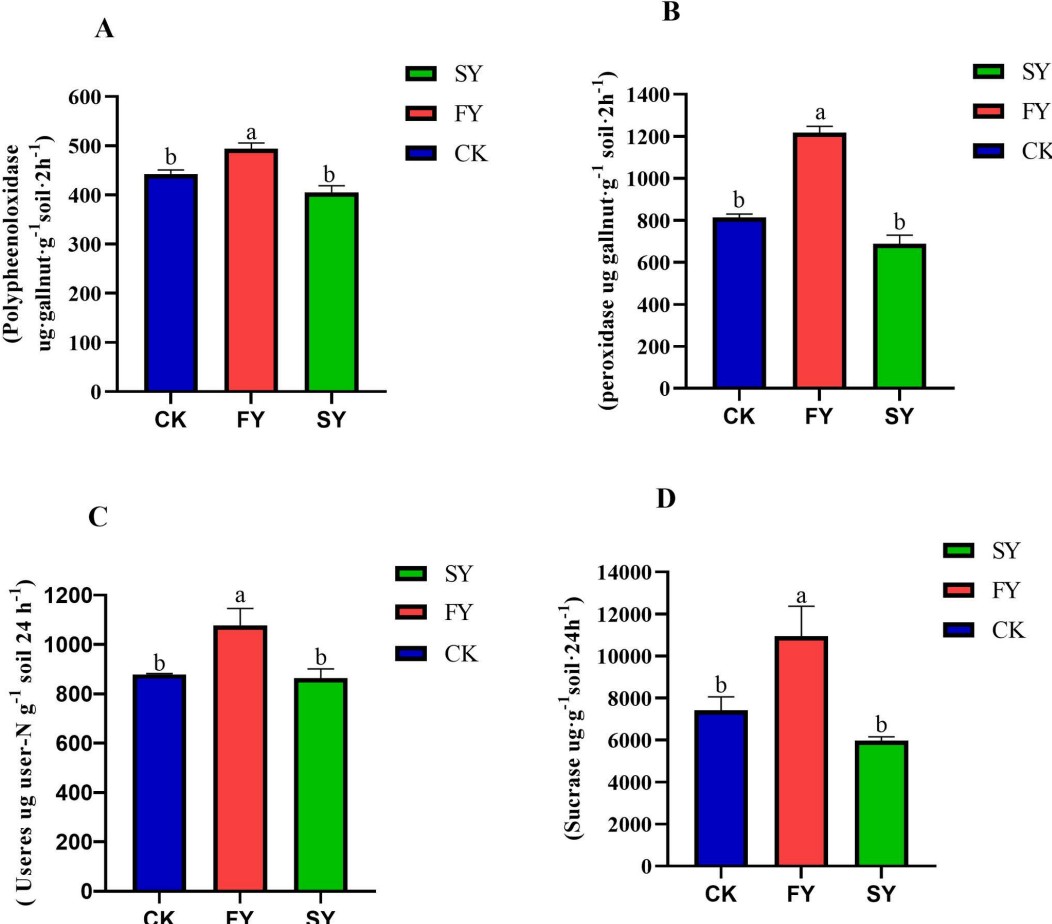

**Fig 1. The effect of continuous monocropping of passion fruit plant on rhizosphere soil enzyme activity.**

**Table 3. Richness and diversity index of bacterial community in the rhizosphere soil of passion fruit under continuous monocropping treatment.**

| Microorganism | Treatment | Observed species index | Shannon index | Simpson index | Chao1index |
|---|---|---|---|---|---|
| Bacteria | CK | 5484a | 10.63a | 0.99a | 6680.29b |
| | FY | 5338b | 10.11b | 0.99a | 7272.08a |
| | SY | 3196c | 8.15c | 0.98b | 4114.66c |

Note: FY (first consecutive monoculture years); SY (second consecutive monoculture years). CK (control soil). Data with different lowercase letters on same column indicate significant different at P < 0.05.

## 3.5. OUT analysis of bacterial communities in the rhizosphere soil of continuously monocroppeed passiflora edulis plants

As illustrated in (Fig 2, S2 Table), the diversity and overlap of operational taxonomic units (OTUs) in bacterial communities of rhizosphere soil from passion fruit plants under continuous monocropping regime were analyzed. The results indicated significant differences and overlaps among specific bacterial OTUs across treatments. Specifically, the OTU levels revealed that in the CK treatment, specific bacteria accounted for 7,007 OTUs (37.29%), while in the FY treatment,

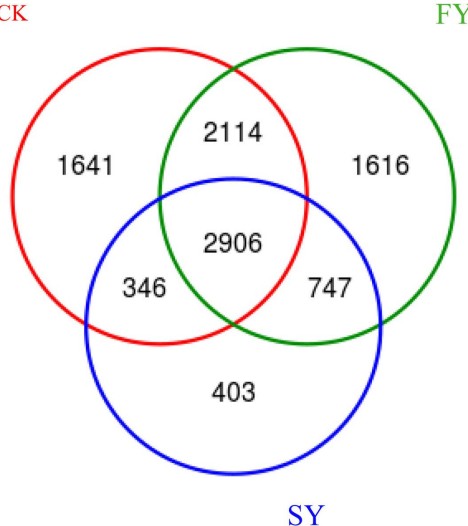

**Fig 2. Wenn diagram of bacterial community in rhizosphere soil of continuous cropping passion fruit.**

specific bacteria accounted for 7,383 OTUs (39.29%), and in the SY treatment, specific bacteria accounted for 4,402 OTUs (23.42%). Additionally, the shared OTU counts were as follows: CK and SY shared 3,252 OTUs (17.31%), FY and SY shared 3,653 OTUs (19.44%), and CK and FY shared 5,020 OTUs (26.71%). Lastly, there were 2,906 OTUs (15.46%) common to CK, FY, and SY treatments. These findings underscore the complexity of bacterial community structures in the rhizosphere soil of passion fruit crop under varying continuous monocropping scenarios, with implications for soil microbial ecology and plant health.

### 3.6. Unifrac heatmap and PCoA analysis of bacterial community in rhizosphere soil of passion fruit crop under continuous monocropping treatment

As depicted in (Fig 3A, S3 Table), bacterial community structures in the rhizosphere soil of passion fruit under FY and CK treatments clustered together, followed by SY treatment, indicating significant differences in bacterial communities between SY and the other two treatments (CK and FY). (Fig 3B, S3 Table) provides further analysis through principal coordinate analysis (PCoA) of the main components PC1 and PC2, which accounted for 64.54% and 27.95% of the variance, respectively. The principal component analysis revealed that bacterial compositions in soil samples from CK, FY, and SY treatments did not belong to the same cluster. However, bacterial compositions in samples from FY and CK treatments were relatively closer to each other. These findings suggest distinct bacterial community structures in the rhizosphere soil of passion fruit plants under different continuous monocropping treatments, which may impact soil health and plant growth.

### 3.7. Analysis of differential bacterial functions and heatmap in rhizosphere soil of passion fruit plants under continuous monocropping regime

As shown in Table 4, at the genus level, when compared with the FY treatment, the relative abundances of *Acidothermus*, *Acidibacter*, *Bacillus*, and *Acidobacterium* in the rhizosphere soil of passion fruit under the SY treatment increased significantly. Meanwhile, the relative abundances of *Rhizomicrobium*, *Nitrospira*, *Burkholderia*, *Sphingomonas*, *Gemmatimonas*, *Streptomyces*, and *Nocardioides* in the SY – treated rhizosphere soil decreased significantly. In the CK treatment, the relative abundances of *Rhizomicrobium*, *Nitrospira*, *Burkholderia*, *Sphingomonas*, *Gemmatimonas*, *Streptomyces*,

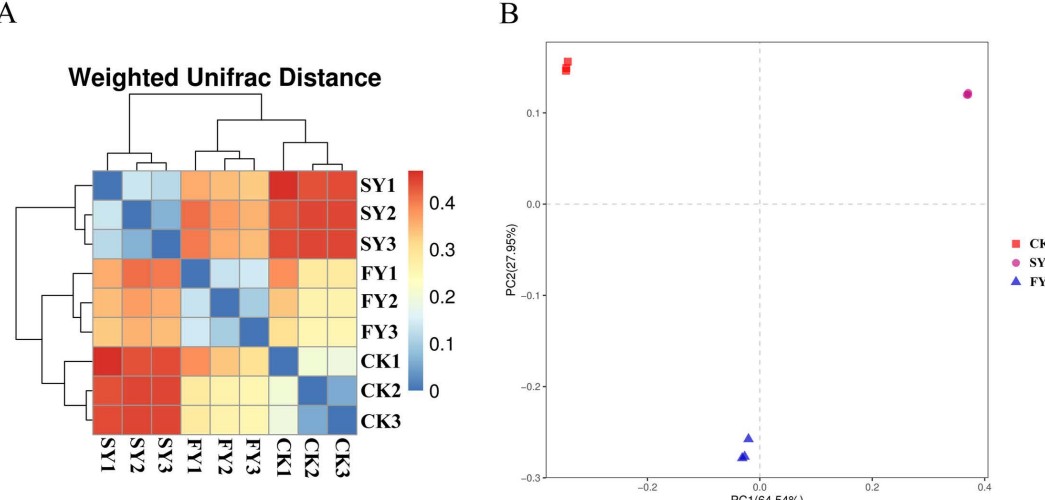

**Fig 3. The weighted UniFrac (A) and PCoA (B) analysis of bacterial samples in passion fruit rhizosphere soil.**

**Table 4. Bacterial community information at the genus level in the rhizosphere soil of continuous monocropping passion fruit plants.**

|  | Genus | Relative abundance/% | | |
| --- | --- | --- | --- | --- |
|  |  | CK | FY | SY |
| Bacteria | *Acidothermus* | 1.80b | 1.55b | 2.50a |
|  | *Acidibacter* | 0.81b | 1.19b | 2.72a |
|  | *Rhizomicrobium* | 1.05b | 2.21a | 1.21b |
|  | *Nitrospira* | 1.47b | 1.94a | 0.79c |
|  | *Burkholderia* | 0.66b | 1.66a | 0.82b |
|  | *Bacillus* | 0.72b | 0.90b | 1.37a |
|  | *Sphingomonas* | 0.62b | 1.84a | 0.20c |
|  | *Acidobacterium* | 0.60b | 0.45b | 1.98a |
|  | *Gemmatimonas* | 0.62b | 1.12a | 0.27c |
|  | *streptomyces* | 0.19b | 1.06a | 0.27b |
|  | *Nocardioides* | 0.47b | 0.81a | 0.14c |

Note: FY (first consecutive monoculture years); SY (second consecutive monoculture years). CK (control soil).Data with different lowercase letters on same column indicate significant different at P < 0.05.

and *Nocardioides* also decreased significantly, and there were no significant differences in other genera. Furthermore, a heatmap analysis of the dominant bacterial communities at the genus level in the rhizosphere soil of continuously – monocropped passion fruit showed that the composition and structure of the bacterial community in the rhizosphere soil changed significantly, as presented in (Fig 4, S4 Table).

### 3.8. Effects on the construction of rhizosphere microbial flora of continuously monocropped passiflora edulis plants

As shown in (Fig 5, S5 Table), in the score of the chessboard| The higher the SES, the stronger the relative contribution of deterministic processes in the assembly process of bacterial communities. Based on the score of the chessboard, it was found that the FY treatment | SES |>2, while the CK and SY treatments | SES |<2, indicating that the rhizosphere

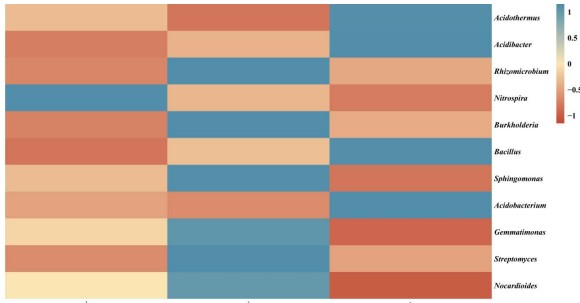

**Fig 4. Analysis of community heatmap at the genus level of dominant bacteria in the rhizosphere soil of continuous cropping passion fruit.**

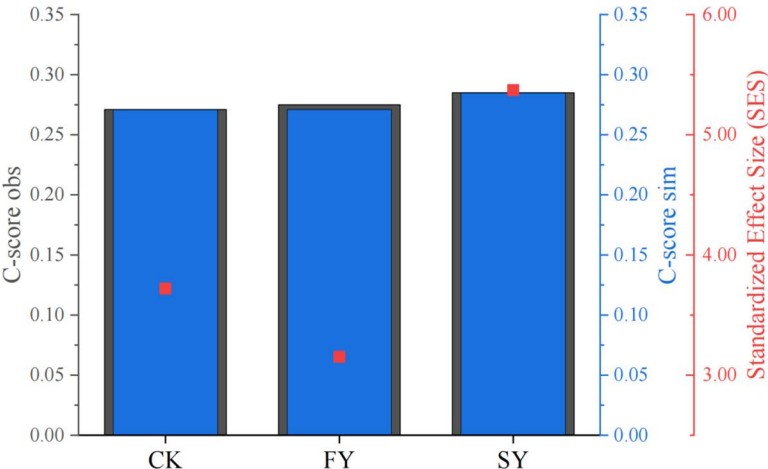

**Fig 5. Chessboard score of rhizosphere soil colonies of continuous cropping passiflora edulis.**

bacteria in the FY continuous cropping of passion fruit dominated the community construction process in a deterministic co-occurrence pattern, while the rhizosphere bacteria in the SY continuous cropping of passion fruit dominated the community construction process in a stochastic co-occurrence pattern, thus transitioning to a deterministic co-occurrence pattern. Further research has confirmed that different soil ecological enhancement techniques play an important role in promoting the evolution and ecological differentiation of rhizosphere bacterial and microbial communities. FY reduces the niche width of dominant bacterial communities in the rhizosphere, indicating relatively abundant nutrient resources, active plant microbe interactions, mutualistic symbiosis, and coordinated development. SY, on the other hand, exhibits the opposite behavior, with different life history strategies for their rhizosphere microbial communities. The former adopts K ecological strategy, while the latter adopts r strategy, implying the instability of their rhizosphere environment. It is speculated that their rhizosphere microbial community adopts r-K life history strategy, indicating the disharmony of SY rhizosphere bacterial environment in continuous cropping of passion fruit, which in turn affects plant root growth and its interaction with microorganisms.

### 3.9. Quantitative analysis of total bacteria and fungi in rhizosphere soil of passion fruit under continuous monocropping treatment

As shown in (Fig 6A, S6 Table), compared with the FY treatment, the total bacterial abundance significantly decreased in both the SY and CK treatments, and there was no significant difference between SY and CK. As can be seen from

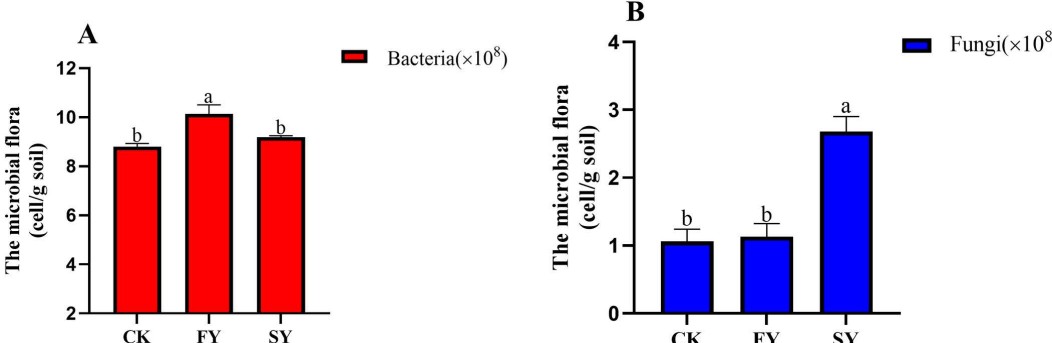

**Fig 6. In situ analysis of total bacteria and fungi in the rhizosphere soil of passion fruit under continuous monocropping treatment.**

(Fig 6B, S6 Table), compared with the FY treatment, the total fungal abundance significantly increased in the SY treatment, while there was no significant difference in the CY treatment. These results indicate that, with the increase of continuous monocropping years, the total number of bacteria in the rhizosphere soil of passion fruit decreases, while the total number of fungi increases.

## 4. Discussion

Continuous cropping obstacles are a common phenomenon in plant cultivation, and the possible causes include soil nutrient imbalance, accumulation of autotoxins, and soil-borne diseases [27]. In recent years, research on plant rhizosphere microbiota has attracted widespread attention from peers both domestically and internationally. The interaction between plants and rhizosphere microorganisms plays a crucial role in the rhizosphere soil ecological environment, plant growth and development, as well as yield and quality. Plant root exudates, as the major regulators of rhizosphere dialogues, significantly influence the rhizosphere soil micro-ecosystem. Extensive studies have shown that continuous monocropping negatively impacts plant growth and development, directly reducing crop yield and quality and increasing plant disease incidence [28]. The current study indicates that continuous monocropping of passion fruit plants significantly increases the contents of total nitrogen, total potassium, total phosphorus, available nitrogen, available phosphorus, available potassium, and organic matter in the FY treatment compared to the SY treatment (Table 1). The pH value was the lowest in the SY treatment. Further analysis revealed significant differences in soil enzyme activities (Fig 1, S1 Table). For instance, the activities of polyphenol oxidase, peroxidase, urease, and invertase in the FY rhizosphere soil increased by 18.0%, 43.6%, 19.8%, and 45.5%, respectively, compared to the SY treatment. These findings suggest that continuous cropping of passion fruit leads to a significant decline in soil fertility, enzyme activity, and acidification. Previous studies have shown that soil enzymes were products of plant roots and soil microorganisms, playing a regulatory role in the transformation and cycling of soil nutrients. Soil enzyme activity can characterize the process and intensity of soil nutrient conversion [29]. HPLC analysis revealed that the content of phenolic acids (syringic acid, vanillin, benzoic acid, ferulic acid) in the rhizosphere soil of continuous cropping passion fruit significantly increased with years, slowing down the decomposition rate of soil organic matter and reducing soil nutrient content, which in turn leads to crop yield reduction [30] (Table 2). Zhou et al. [19] reported that phenolic acids in continuous cropping cucumber root exudates can inhibit the ion absorption by cucumber roots, with the concentration of phenolic acids and the pH value of the medium being the two main factors. A decrease in pH increases the inhibitory effect of phenolic acids on ion absorption by cucumber roots. This indicates that phenolic acids were more likely to exhibit allelopathic effects in acidic soils, which is consistent with the researchers findings. These results suggest that phenolic acid substances have allelopathic effects on most plants.

Through chessboard score analysis, this study found that there is a positive feedback effect that can promote the rhizosphere microecological effects of continuous monocropping of passion fruit plants, especially with a significant increase in the abundance of certain beneficial bacterial communities. At the genus level, the relative abundance of Acidothermus, Acidobacterium, and Acidobacterium in the rhizosphere soil of continuous monocropping of passion fruit plants significantly increased. This further indicates soil acidification with increasing years of passion fruit cultivation [31]. In the FY rhizosphere soil, the relative abundances of *Rhizomicrobium*, *Nitrospira*, *Burkholderia*, and *Streptomyces* significantly increased (Table 4). *Burkholderia* can inhibit the growth of pathogens and effectively serve as a bio-organic fertilizer to improve plant growth [32]. *Nitrospira* plays a crucial role in the second step of the nitrogen cycle by oxidizing nitrite [33,34]. *Streptomyces* can act as PGPR (Plant Growth-Promoting Rhizobacteria) to reduce plant diseases and is associated with the inhibition of many soil-borne plant diseases [35]. The qRT-PCR analysis revealed that with increasing years, the total bacterial count decreased, while the total fungal count increased in the rhizosphere soil of continuous cropping passion fruit (Fig 6, S6 Table). Wu et al. [36] reported that continuous cropping disrupts the micro-ecosystem of the rhizosphere soil, promoting the transformation from a bacterial-dominated to a fungal-dominated soil, with an increase in pathogenic fungi and a decrease in beneficial bacteria, leading to aggravated diseases. This result was consistent with previous findings. These findings collectively underscore the detrimental effects of continuous monocropping on soil microbial communities and plant health.

## 5. Conclusion

This study demonstrated that with increasing years of continuous monocropping, the enzyme activities and pH of the rhizosphere soil of passion fruit plants decreased, while the content of phenolic acids increased. These changes led to significant alterations in the total bacterial and fungal populations, resulting in an imbalance in the rhizosphere soil microecology and indirectly causing continuous monocropping obstacles in passion fruit plants. However, the mechanism by which allelochemicals in the rhizosphere soil of continuously monocropped passion fruit plants mediated the relationships between soil pH, phenolic acids, and microorganisms in the rhizosphere ecology of continuous cropping obstacles still needed to be further elucidated through designed experiments.

## Supporting information

**S1 Table. The content of rhizosphere soil enzymes in passion fruit with different growth years.**
(XLS)

**S2 Table. The differences in OTU composition of rhizosphere soil bacterial communities of passion fruit with different growth years.**
(XLSX)

**S3 Table. The differences in the structural composition of rhizosphere soil bacterial communities of passion fruit with different growth years.**
(XLSX)

**S4 Table. The heat map analysis of rhizosphere soil dominant bacterial communities at the genus level in passion fruit with different growth years.**
(XLS)

**S5 Table. The chessboard scoring of rhizosphere soil bacterial colonies in passion fruit with different growth years.**
(XLSX)

**S6 Table. The differences in total abundance of bacteria and fungi in rhizosphere soil of passion fruit with different growth years.**
(XLS)

## Author contributions

**Data curation:** Weiwei Lin, Zhihan Chen.

**Formal analysis:** Weiwei Lin, Zhihan Chen, Zhaowei Li.

**Funding acquisition:** Wenxiong Lin.

**Project administration:** Wenxiong Lin, Zhaowei Li.

**Writing – original draft:** Weiwei Lin, Zhihan Chen.

**Writing – review & editing:** Wenxiong Lin.

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
