## [Decision Letter · Decision Letter 0]

PONE-D-24-60574Changes in Rhizosphere Enzyme Activity and Microbial Community of Continuously Cultivated Passiflora edulisPLOS ONE

Dear Dr. Lin,

Thank you for submitting your manuscript to PLOS ONE. After careful consideration, we feel that it has merit but does not fully meet PLOS ONE’s publication criteria as it currently stands. Therefore, we invite you to submit a revised version of the manuscript that addresses the points raised during the review process.

We look forward to receiving your revised manuscript.

Kind regards,

Abhijeet Shankar Kashyap

Academic Editor

PLOS ONE

Journal Requirements:

“Supported by the Natural Science Foundation of Fujian Province (2022J01142), the School-level Project of Zhangzhou Institute of Technology in 2024 (zzykyk24004), and the Doctoral Research Start-up Fund Project of Zhangzhou Institute of Technology in 2024 (Project No.: ZZYB2403)”

“Supported by the Natural Science Foundation of Fujian Province (2022J01142), the School-level Project of Zhangzhou Institute of Technology in 2024 (zzykyk24004), and the Doctoral Research Start-up Fund Project of Zhangzhou Institute of Technology in 2024 (Project No.: ZZYB2403)”

“Supported by the Natural Science Foundation of Fujian Province (2022J01142), the School-level Project of Zhangzhou Institute of Technology in 2024 (zzykyk24004), and the Doctoral Research Start-up Fund Project of Zhangzhou Institute of Technology in 2024 (Project No.: ZZYB2403)”

5. We note that your Data Availability Statement is currently as follows: [All relevant data are within the manuscript and its Supporting Information files.]

6. PLOS requires an ORCID iD for the corresponding author in Editorial Manager on papers submitted after December 6th, 2016. Please ensure that you have an ORCID iD and that it is validated in Editorial Manager. To do this, go to ‘Update my Information’ (in the upper left-hand corner of the main menu), and click on the Fetch/Validate link next to the ORCID field. This will take you to the ORCID site and allow you to create a new iD or authenticate a pre-existing iD in Editorial Manager.

Additional Editor Comments:

The manuscript of “Changes in Rhizosphere Enzyme Activity and Microbial Community of Continuously Cultivated Passiflora edulis” did many works. The manuscript is interesting and systematic, but there were many questions existed in the present study. Authors may incorporate all suggestion and details asked by the reviewers.

Reviewers' comments:

Reviewer's Responses to Questions

**Comments to the Author**

1. Is the manuscript technically sound, and do the data support the conclusions?

Reviewer #1: Yes

Reviewer #2: Partly

Reviewer #3: Partly

2. Has the statistical analysis been performed appropriately and rigorously? 

Reviewer #1: Yes

Reviewer #2: Yes

Reviewer #3: Yes

3. Have the authors made all data underlying the findings in their manuscript fully available?

Reviewer #1: Yes

Reviewer #2: Yes

Reviewer #3: Yes

4. Is the manuscript presented in an intelligible fashion and written in standard English?

Reviewer #1: Yes

Reviewer #2: No

Reviewer #3: No

5. Review Comments to the Author

Reviewer #1: Minor points

1- In abstract, 14line and 18line: You wrote (We 2 times)! The rule of manuscript

writing is to avoid using (We). So you should delete (We) and use formal

scientific words (This study or The current study or The present study).

2- After abstract; you should write Key words of your manuscript, so why you did

not write them!!!

3- In introduction, the rule of manuscript writing is to write the aims of your study

in the end of introduction part. Kindly add the aims.

4- In 371line: You wrote discuss!!! Kindly convert it to discussion.

5- In discussion, 382line, 404line and 425line: You wrote (Our 3 times)! The rule

of manuscript writing is to avoid using (Our). So you should delete (Our), use the

formal scientific words (This study or The current study or The present

study).

Kind regards

Reviewer #2: This manuscript addresses a topic of significant practical relevance, shedding light on potential causes of continuous cropping obstacles in Passiflora edulis. However, the manuscript needs substantial revisions in terms of writing quality and data presentation. The data analysis is quite basic, and the figures and tables lack scientific rigor and proper formatting.

There are several issues with the abstract. For instance, in line 37, the mention of beneficial microorganisms is not supported by the data presented, making that statement questionable.

The writing in the introduction does not meet the standards of scientific discourse and requires significant revisions to adhere to proper grammar and academic style. Several instances suggest a lack of attention to detail in the writing process, and I encourage the authors to take more care in refining these sections. Specific issues include:

Line 47-49: The font changes in these lines.

Line 63: Incorrect punctuation.

Line 63-65: The conclusion here is incorrect and lacks appropriate citations. The source of this claim is unclear.

Line 67-69: Grammar issues.

Line 82-85: The meaning of this section is unclear—either punctuation is missing, or there is a grammatical error. Similar small issues appear throughout the manuscript.

The Materials and Methods section contains several errors, and I strongly recommend that the authors carefully review this section. In particular, the enzyme activity measurement method should be described in more detail.

The inclusion of a control (CK) seems unnecessary, as the primary focus of the study is the comparison between the first and second years of continuous cropping.

Furthermore, the enzyme activity measurement methods are repeated in lines 124 and 127. These should be consolidated for clarity.

Line 102: Grammar error.

Line 139: The method for obtaining rhizosphere soil is not explained. This needs to be clarified.

The discussion section is underdeveloped, and the presentation of the figures and tables is not sufficiently polished.

Reviewer #3: The manuscript of “Changes in Rhizosphere Enzyme Activity and Microbial Community of Continuously Cultivated Passiflora edulis” did many works. The manuscript is interesting and systematic, but there were many questions existed in the present study.

1.There were many grammar mistakes involved in this manuscript, including abstract and more. I suggest that authors polish the language for this study, such as in the Abstract, line 23-35, using words like “significant changes, significant decreased, increased and more”, please supply compared with what treatment, or what indexes?

2.This manuscript lack of keywords, please carefully check and revise.

3.Line 44-48, line 59-61, I wonder if there was a format mistake in these sentence.

4.I suggest that authors supply more information in the introduction for the planting history, planting question, planting methods of passion fruit, but these origin history.

5.Line 65, please check this reference, I wonder that if this reference is correct or wrong.

6.In the introduction, material and methods, and discussion, there were many wrong format references, please carefully check and revise.

7.I suggest that author supply a paragraph in the introduction to introduce science question of this study.

8.Results was too long and boring, please carefully check and simplify.

9.The discussion need to be completed, I suggest that author in-depth search of relevant literature, careful summary, and comparative analysis with the results of this study, to find the reasons.

10.I want to see the ANOSIM test for the PCoA result.

6. PLOS authors have the option to publish the peer review history of their article (what does this mean? ). If published, this will include your full peer review and any attached files.

**Do you want your identity to be public for this peer review?** For information about this choice, including consent withdrawal, please see our Privacy Policy .

Reviewer #1: No

Reviewer #2: No

Reviewer #3: No

---

## [Author Response · Author response to Decision Letter 1]

24 Mar 2025

We have written a response letter to address the comments raised by the editor and reviewers.The file name is: Response to Reviewers - 0307

---

## [Decision Letter · Decision Letter 1]

PONE-D-24-60574R1Changes in Rhizosphere Enzyme Activity and Microbial Community of Continuously Cultivated Passiflora edulisPLOS ONE

Dear Dr. Lin,

Thank you for submitting your manuscript to PLOS ONE. After careful consideration, we feel that it has merit but does not fully meet PLOS ONE’s publication criteria as it currently stands. Therefore, we invite you to submit a revised version of the manuscript that addresses the points raised during the review process.Incorporate Reviewer 2 suggestions.

We look forward to receiving your revised manuscript.

Kind regards,

Abhijeet Shankar Kashyap

Academic Editor

PLOS ONE

Journal Requirements:

Additional Editor Comments :

Incorporate Reviewer 2 suggestions.

Reviewers' comments:

Reviewer's Responses to Questions

**Comments to the Author**

1. If the authors have adequately addressed your comments raised in a previous round of review and you feel that this manuscript is now acceptable for publication, you may indicate that here to bypass the “Comments to the Author” section, enter your conflict of interest statement in the “Confidential to Editor” section, and submit your "Accept" recommendation.

Reviewer #1: All comments have been addressed

Reviewer #2: All comments have been addressed

2. Is the manuscript technically sound, and do the data support the conclusions?

Reviewer #1: Yes

Reviewer #2: Partly

3. Has the statistical analysis been performed appropriately and rigorously? 

Reviewer #1: Yes

Reviewer #2: Yes

4. Have the authors made all data underlying the findings in their manuscript fully available?

Reviewer #1: Yes

Reviewer #2: Yes

5. Is the manuscript presented in an intelligible fashion and written in standard English?

Reviewer #1: Yes

Reviewer #2: No

6. Review Comments to the Author

Reviewer #1: (No Response)

Reviewer #2: After thoroughly reviewing your submission, I sincerely apologize that I was unable to locate your Response despite my best efforts to examine all the links provided. As the required content remains inaccessible, I must request the author to continue revising this manuscript.

7. PLOS authors have the option to publish the peer review history of their article (what does this mean? ). If published, this will include your full peer review and any attached files.

**Do you want your identity to be public for this peer review?** For information about this choice, including consent withdrawal, please see our Privacy Policy .

Reviewer #1: No

Reviewer #2: No

---

## [Author Response · Author response to Decision Letter 2]

16 May 2025

The specific comments raised by the reviewers and the editor have been clearly responded to in the 'Response to Reviewers'.

---

## [Decision Letter · Decision Letter 2]

PONE-D-24-60574R2Changes in Enzyme Activity and Microbial Community of Rhizosphere Soil under Continuously Monocultured  Passiflora edulis  TreatmentPLOS ONE

Dear Dr. Lin,

Thank you for submitting your manuscript to PLOS ONE. After careful consideration, we feel that it has merit but does not fully meet PLOS ONE’s publication criteria as it currently stands. Therefore, we invite you to submit a revised version of the manuscript that addresses the points raised during the review process.

We look forward to receiving your revised manuscript.

Kind regards,

Abhijeet Shankar Kashyap

Academic Editor

PLOS ONE

Journal Requirements:

Reviewers' comments:

Reviewer's Responses to Questions

**Comments to the Author**

1. If the authors have adequately addressed your comments raised in a previous round of review and you feel that this manuscript is now acceptable for publication, you may indicate that here to bypass the “Comments to the Author” section, enter your conflict of interest statement in the “Confidential to Editor” section, and submit your "Accept" recommendation.

Reviewer #2: All comments have been addressed

2. Is the manuscript technically sound, and do the data support the conclusions?

Reviewer #2: Partly

3. Has the statistical analysis been performed appropriately and rigorously? 

Reviewer #2: No

4. Have the authors made all data underlying the findings in their manuscript fully available?

Reviewer #2: No

5. Is the manuscript presented in an intelligible fashion and written in standard English?

Reviewer #2: Yes

6. Review Comments to the Author

Reviewer #2: I believe the authors still need to address several issues, such as standardizing the formatting throughout the manuscript and submitting the sequencing data to a public database (e.g., the National Genomics Data Center).

7. PLOS authors have the option to publish the peer review history of their article (what does this mean? ). If published, this will include your full peer review and any attached files.

**Do you want your identity to be public for this peer review?** For information about this choice, including consent withdrawal, please see our Privacy Policy .

Reviewer #2: No

---

## [Author Response · Author response to Decision Letter 3]

29 Jun 2025

Dear Editor

We appreciate the reviewer’s comments on standardizing the manuscript formatting and submitting sequencing data.

All formatting inconsistencies throughout the manuscript (including referencing styles, figure/table captions, and text spacing) have been carefully revised to adhere to PLOS ONE’s guidelines.

The required sequencing data have been deposited in the National Center for Biotechnology Information (NCBI) under the accession number [PRJNA1280495]. The detailed data can be accessed via the following link: [https://dataview.ncbi.nlm.nih.gov/object/PRJNA1280495?reviewer=i1ut7ml58v5vpt2bdq41qqfogb].

These revisions ensure compliance with the journal’s standards and enhance the transparency and reproducibility of the study. We thank the reviewer for prompting these improvements.

Sincerely

Weiwei Lin

---

## [Editor Report · Decision Letter 3]

Changes in Enzyme Activity and Microbial Community of Rhizosphere Soil under Continuously Monocultured  Passiflora edulis  Treatment

PONE-D-24-60574R3

Dear Dr. Lin,

We’re pleased to inform you that your manuscript has been judged scientifically suitable for publication and will be formally accepted for publication once it meets all outstanding technical requirements.

Kind regards,

Abhijeet Shankar Kashyap

Academic Editor

PLOS ONE
---

## [Editor Report · Acceptance letter]

PONE-D-24-60574R3

PLOS ONE

Dear Dr. Lin,

I'm pleased to inform you that your manuscript has been deemed suitable for publication in PLOS ONE. Congratulations! Your manuscript is now being handed over to our production team.

Kind regards,

on behalf of

Dr. Abhijeet Shankar Kashyap

Academic Editor

PLOS ONE